# GAUSSIAN PROCESS-BASED CORRUPTION-RESILIENCE FORECASTING MODELS

## ABSTRACT

Time series forecasting is a challenging due to complex temporal dependencies and unobserved external factors, which can lead to incorrect predictions by even the best forecasting models. Using more training data is one way to improve the accuracy, but this source is often limited. In contrast, we are building on successful denoising approaches for image generation. When a time series is corrupted by the common isotropic Gaussian noise, it yields unnaturally behaving time series. To avoid generating unnaturally behaving time series that do not represent the true error mode in modern forecasting models, we propose to employ Gaussian Processes to generate smoothly-correlated corrupted time series. However, instead of directly corrupting the training data, we propose a joint forecast-corrupt-denoise model to encourage the forecasting model to focus on accurately predicting coarse-grained behavior, while the denoising model focuses on capturing fine-grained behavior. All three parts are interacting via a corruption model which enforces the model to be resilient. Our extensive experiments demonstrate that our proposed approach is able to improve the forecasting accuracy of several state-of-the-art forecasting models as well as several other denoising approaches. The code for reproducing our main result is open-sourced and available online.[1]

## 1 INTRODUCTION

Time series forecasting is a vital foundational technology in many important domains such as in economics Capistrán et al. (2010), health care Lim (2018), demand forecasting Salinas et al. (2020) and autonomous driving Chang et al. (2019). Despite the recent advances in neural networks, time series forecasting still remains a challenging problem due to the complex and dynamic temporal dependencies across different time scales. Furthermore, the existence of hidden external factors is a challenge for forecasting models to correctly mimic the temporal behavior of the variable of interest. Thus, developing creative approaches to improve the model accuracy and resilience without adding more training data is important to overcome these challenges.

Denoising models Ho et al. (2020) have recently gained popularity in generating high quality images. Typically denoising models Vincent et al. (2010) are trained to reverse an image corrupting process, thereby learning correlation patterns of the application domain. Including weak supervision via a denoising objective will train models to simultaneously predict and denoise, resulting in more resilient models overall.

In this work, we rethink ideas from variational denoising models in applications to time series forecasting task, which is defined as follows:

**Time series forecasting task.** Given $\kappa$ time series observations prior to cutoff a time step $t_0$, the task is to predict values of the target variable $\gamma$ for the next $\tau$ time steps into the future (from $t_0$ to $t_0 + \tau$). We refer to the given time series as $X = \{\boldsymbol{x}_t; \boldsymbol{\gamma_t}\}_{t=t_0-\kappa}^{t_0}$, and the to be predicted series as $Y = \{\boldsymbol{\gamma}_t\}_{t=t_0}^{t_0+\tau}$. The target variable can be multivariate with $\boldsymbol{\gamma}_t \in \mathbb{R}^{d_y}$, although we focus on datasets with univariate target variables ($d_y = 1$). The given time series observations $X$ include $d_x$ covariates, such as season or time of day. Additionally, each observation $X_t$ encompasses the target variable $\boldsymbol{\gamma}_t$ prior to time step $t_0$, yielding $X_t \in \mathbb{R}^{d_x+d_y}$.

---

[1]Code available at `https://anonymous.4open.science/r/Corruption-resilient-Forecasting-Models-15E8`

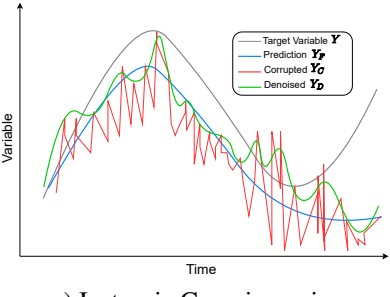
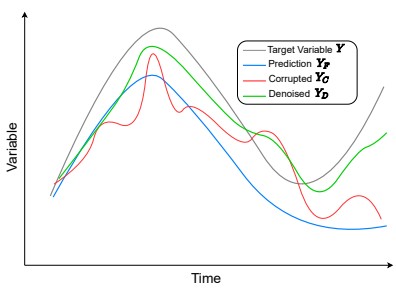

a) Isotropic Gaussian noise        b) Gaussian Process noise

Figure 1: A synthetic example of corrupting and denoising the prediction. The left figure a) illustrates that corrupting and denoising the prediction with isotropic Gaussian noise results in less desirable forecasts, where the denoising model attempts to remove the jitters. However, as depicted in the right figure b) corrupting and denoising the prediction with our proposed Gaussian Process model results in a smooth behavior with improved fine-grained details.

**Idea.** When applied to image generation, denoising models usually revert a corruption process of pixel-wise isotropic Gaussian noise. With application to time series forecasting, however, this isotropic corruption will introduce what we call *jitters* (see red function in Figure 1 left), as the noise added to one time step is independent of noise added to subsequent time steps. Since most time series exhibit smooth behavior, the result of the isotropic corruption process yields time series that are rather unnaturally behaving. Moreover, according to our preliminary experiments, most of the state-of-the-art forecasting models do not produce predictions with many jitters, hence the isotropic corruption model is not representing the typical error modes of modern forecasting models. Hence, we hypothesize that the benefit of isotropic corruption is somewhat limited.

In this work we explore alternative corruption processes for training denoising models for time series forecasting. The goal is to train a corruption model that is most beneficial for improving performance in the time series domain. The typically erroneous predictions are usually of smooth, yet incorrect temporal behavior. As our goal is to train a denoiser to correct forecasting mistakes, we are interested in a corruption model that generates such smooth, yet faulty temporal behavior. As the isotropic Gaussian will generate jitters due to its temporally uncorrelated nature, in line with Robinson et al. Robinson et al. (2018) we will employ a Gaussian Process that naturally models correlation across time to provide smooth functions.

**Related work.** Our goal is to improve the forecasting ability of the existing time series forecasting models. Among various time series forecasting models, the ones that utilize transformers have demonstrated superior performance Li et al. (2019); Fan et al. (2019). But even the state-of-the-art time series forecasting models make wrong predictions. We will apply our approach to improve two of the best forecasting models, the Autoformer and the Informer model. The Autoformer model Wu et al. (2021) improves on the basic attention mechanism by decomposing a time series into sub-series and incorporating an auto-correlation mechanism to capture the correlation between these sub-series. This leads to gains in efficiency and accuracy. The Informer model Zhou et al. (2021) employs ProbSparse attention, which prunes the attention matrix by focusing on samples that are outliers from a uniform distribution. Both these models are included in our experimental evaluation.

Probabilistic time series forecasting models such as DeepAR Salinas et al. (2020) generate predictions by drawing samples from a learned Isotropic Gaussian distribution. These models aim to learn the parameters of the Gaussian distribution via a deep neural network model such as LSTMs Hochreiter & Schmidhuber (1997).

Denoising models often revert a corruption process applied to the input during training to increase the robustness and generalization of the model. Therefore, the performance of time series forecasting models could as well be improved when integrated with a denoising approach. One of such approaches is the TimeGrad Rasul et al. (2021) forecasting model that uses denoising diffusion models to reverse the isotropic Gaussian noise added to the input time series. TimeGrad estimates the parameters of the Gaussian distribution using recurrent neural networks architectures LSTM/GRU. However, prior

works including Koohfar & Dietz (2022); Khandelwal et al. (2018) show that LSTMs do not perform competitively compared to transformer-based forecasting methods.

DLinear model proposed by Zeng et al Zeng et al. (2022) proposes a *navive* time series forecasting model that only uses simple multi-layer perceptrons projections from previous observations to make predictions. We compare our proposed model to DLinear model in our experimental section.

Recently, Li et al Li et al. (2022) introduce D3VAE, a novel approach that combines bidirectional variational auto-encoder techniques with diffusion, denoising, and disentanglement to enhance time series representation. However, in line with DLinear, this approach does not include sequential modeling techniques, resulting in limited competitiveness when compared to transformer-based forecasting methods.

**Contributions.** In this work, we take a different approach by enhancing the performance of forecasting models through a corruption-resilient forecasting framework. (1) Rather than corrupting the input during training, we advocate for an end-to-end forecast-corrupt-denoise framework that encourages a separation of concerns for the forecasting and denoise models. (2) Our complementary approach can be readily added to a wide-range of forecasting models. (3) Our novel approach is an alternative towards approaches that use denoising solely during training or when leveraging boosting. We experimentally show that our approach predominately outperforms many of those baselines. (4) We further demonstrate that for smooth time series data, isotropic Gaussians are not a suitable corruption model.

## 2 METHODOLOGY

Time series forecasting models using a ground truth series $Y$, are trained to learn the expected behavior of the target variable over time. However, complex dependencies and unobserved external factors can lead to erroneous predictions. Adding more training data can help to improve the forecasting model's performance, but it is often a limited resource. In this work we explore how ideas from denoising models can increase the generalization of the family of forecasting models.

### 2.1 BACKGROUND: ISOTROPIC DENOISING APPROACHES

Denoising models are trained to reverse a corruption process. For this purpose, true data $X$ is corrupted with noise to obtain a corrupted version $\tilde{X}$. The denoising part of the model is trained to, given $\tilde{X}$, predict the original data $X$. Often this is modelled by predicting the noise, which is then subtracted from $\tilde{X}$ to obtain the original data $X$. Denoising models can be trained for the iterative reversal of noise (reasoning on a series of latents with different noise level) or in single step with a conditional model $X|\tilde{X}$ Deja et al. (2022). The denoising objective can be optimized by itself or in combination with other objectives such as prompted image generation or time series forecasts Nichol & Dhariwal (2021).

Most current work uses a simple corruption process that employs an isotropic Gaussian noise model, where $\tilde{X}|X \sim X + \mathcal{N}(0, \sigma^2 \mathbf{I})$ that acts independently on different data points, i.e., pixels for image generation Nichol & Dhariwal (2021) or time steps for time series forecasts Rasul et al. (2021).

### 2.2 ISSUES WITH UNCORRELATED NOISE MODELS FOR TIME SERIES

Isotropic Gaussian noise is one of the most commonly employed corruption processes, which provides noise that is identically and independently distributed (*i.i.d.*) and when used to generate observations for time series are lacking smooth behavior over time. While isotropic Gaussian noise corruption can (and are) applied to time series, the result is a corruption in the form of jitters (as depicted in red in Figure 1a). The effect on the denoising problem is that the denoiser may only learn to remove jitters. However, time series forecasting models are based on the assumption that data points are correlated over time—and hence not i.i.d. The effect on the training problem is that the denoiser may only learn to remove jitters. However, most errors in forecasting models are not due to jitters, as predicted forecasts are usually smooth functions that merely exhibit incorrect behavior. In this work we go even one step further and train the corruption model to match typical error modes of the underlying forecasting model.

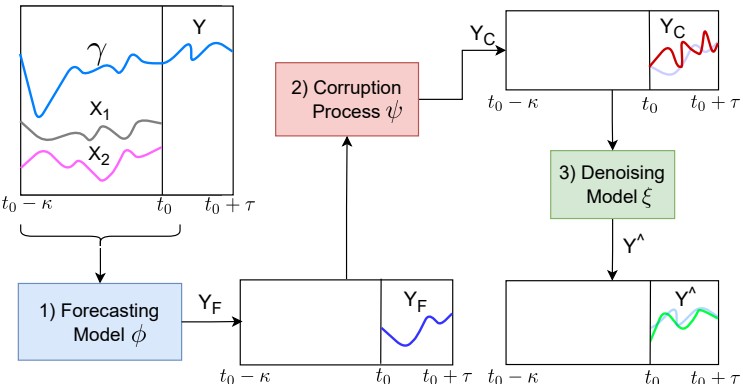

Figure 2: Our proposed model framework for end-to-end training the forecasting and denoising model. The multi-step neural network forecasting model predicts $Y_F$ from covariates $X_1$, $X_2$, and previous target variable $\gamma$. The predictions $Y_F$ are then corrupted by our GP model to obtain $Y_C$. The corrupted predictions $Y_C$ are then denoised by our denoising model to obtain $Y_D = \hat{Y}$.

## 2.3 SMOOTH CORRUPTION WITH GAUSSIAN PROCESSES

To obtain a more resilient model, we instead propose to use a corruption model that generates smooth temporally-correlated functions, to train the denoising model, as depicted in Figure 1b). We employ a Gaussian Process (GP) which models the correlation between consecutive samples in a sequence of observations via a kernel function $k_\psi$.

We replace the isotropic Gaussian corruption model with a corruptive Gaussian Process (GP) $c_\psi$ as follows:

$$c_\psi(\tilde{X}|X) \sim \mathcal{N}(\tilde{X}; X, k_\psi(X, X) + \sigma^2 \mathbf{I}) \tag{1}$$

where $\psi$ denotes the parameters of the GP kernel.

In Section 3 we will experimentally support our claim that corruption with GPs leads to more effective training for time series forecasting than corruption with isotropic Gaussians.

## 2.4 CORRUPTION-RESILIENT FORECASTING FRAMEWORK

There are many ways to exploit ideas from denoising models for time series forecasting: as separate negative training data, as a secondary objective on true time series, via auto-encoders, or as an integral part of an end-to-end model. Based on preliminary experiments, in this work we integrate ideas into a joint forecast-corrupt-denoise model. The model integrates 1) a forecasting model, 2) a corruption process, and 3) a denoising model as depicted in Figure 2. All parts are jointly trained for best MSE performance. We describe these components in detail below.

**1) Forecasting model:** Any time series forecasting model can be used here that, given the observations $X = \{x_t; \gamma_t\}_{t=t_0-\tau}^{t_0}$ predicts the future target variables $Y_F$, as represented by the blue box in Figure 2. We experimentally demonstrate that our will help train more accurate and resilient forecasting models. We refer to the set of parameters of the forecasting model as $\phi$.

**2) Corruption process:** The initial predictions $Y_F$ are corrupted with a noise function $c$, depicted as a light red box in Figure 2. As described above, we suggest to use a Gaussian Process as corruption model to obtain $Y_C \sim \mathcal{N}(Y_C; Y_F, k_\psi(Y_F, Y_F) + \sigma^2 \mathbf{I})$. GP parameters $\psi$ are trained jointly with other parameters of our end-to-end forecasting and denoising model. Alternative corruption processes could be used here, which we explore in Section 3.

**3) Denoising Model:** Given the corrupted predictions $Y_C$, the denoising model aims to revert the corruption process and improve the initial forecasting. While many architectures could be chosen for the denoising model, we choose to use the same time series forecasting model with a new set of parameters $\xi$ as the denoiser to obtain final predictions $Y_D = \hat{Y}$. By

employing the time series forecasting model as the denoiser, the denoising model is able to eliminate the corruption from the corrupted forecasts while retaining crucial details and patterns in the time series data acquired through the set of parameters $\xi$. The denoising model is represented by the green box in Figure 2.

The result is a compound model that encourages the initial forecasting model to focus on modelling coarse-grained behavior, and a denoising model that corrects the fine-grained details. This is encouraged by a corruption process that will "blur" fine-grained details in the forecast, and a denoising model that focuses on correcting these fine-grained details. Additionally, the denoising component acts as a fall-back for when the initial forecasting model fails, reducing the likelihood of catastrophic errors.

Note that for fixed training data $X$ and $Y$, a new corruption is sampled in every epoch, deterring the model from overfitting to any particular corruption.

To optimize the GP model's parameters, we employ a strategy reminiscent of the scalable variational Gaussian Process (GP) method introduced by Hensman et al. in Hensman et al. (2015). Their scalable variational GP technique offers a computationally efficient approximation of the GP model, achieving nearly linear computational complexity for $k_\psi(X, X)$ as the forecasting horizon increases.

## 2.5 END-TO-END FORECASTING AND GP LOSS

With an abundance of training data, the compound model could be trained end-to-end, predicting $\hat{Y}$ from given $X$ and minimizing the distance to the ground truth $Y$ via an MSE (or $L_2$) forecasting loss.

We optimize the parameters of our GP model using the ground truth $Y$. This allows for the efficient optimization of the parameters of variational Gaussian processes. The compound loss function employed for end-to-end training is defined as follows, where the variational evidence lower bound (ELBO) optimizes the corruption model to obtain an ideal noise process:

$$\mathcal{L} = \underbrace{L_{\text{MSE}}}_{\text{forecasting loss}} (\hat{Y} = Y | X) + \underbrace{\lambda L_{\text{ELBO}}}_{\text{GP loss}}(Y_C = Y | Y_F) \tag{2}$$

In our experiments, following Nichol et al Nichol & Dhariwal (2021) we set $\lambda$ to a small number ($\lambda = 0.001$) to prevent the loss $L_{\text{ELBO}}$ from overwhelming the $L_{\text{MSE}}$ loss. Figure 2 illustrates our end-to-end framework.

## 3 EXPERIMENTS

We experimentally demonstrate the success of our forecast-corrupt-denoise approach across three datasets and two state-of-the-art forecasting models. We first focus on demonstrating the efficacy of our treatment and the importance of using correlated noise of GPs, rather than isotropic Gaussians. Next, we compare our denoising approach to a wide range of canonical denoising and ensemble baselines.

## 3.1 EXPERIMENTAL SETUP

We lay out the conditions for our experimental evaluation.

**Datasets.**   We select three widely used datasets that have been used for training and validation by a significant amount of research papers Salinas et al. (2020); Li et al. (2019); Wu et al. (2021); Zhou et al. (2021).

**Traffic** [2]: A univariate dataset, containing the occupancy rate ($y_t \in [0, 1]$) of 440 SF Bay Area freeways, aggregated on hourly interval.

---

[2]Traffic `https://archive.ics.uci.edu/ml/machine-learning-databases/00204/PEMS-SF.zip`

Table 1: Overall results of the quantitative evaluation of corruption-resilient and baseline forecasting models in terms of **MSE**. We compare the forecasting models on all three datasets with different number of forecasting steps. A lower **MSE** indicates a better model. In all cases our forecast-corrupt-denoise approach with GPs improves performance of the original forecasting model (Autoformer) and isotropic Gaussian noise model (AutoDI). (Note that to provide a fair comparison, all the baseline models considered in this study were trained and evaluated under the *same* experimental setup as our proposed model. Consequently, the reported results may differ from those originally reported in the respective baseline papers. We provide the baseline models implementation in our online repository).

| Dataset | Horizon | AutoDG(Ours) | Autoformer | AutoDI | NBeats | DLinear | DeepAR | CMGP | ARIMA |
|---|---|---|---|---|---|---|---|---|---|
| Traffic | 24 | **0.392** ±0.006 | 0.412 ±0.006 | 0.405 ±0.003 | 0.475 ±0.008 | 0.553 ±0.000 | 0.888 ±0.000 | 0.824 ±0.000 | 1.436 ±0.000 |
| | 48 | **0.387** ±0.001 | 0.422 ±0.004 | 0.416 ±0.001 | 0.462 ±0.012 | 0.547 ±0.000 | 0.944 ±0.000 | 0.828 ±0.000 | 1.444 ±0.000 |
| | 72 | **0.380** ±0.001 | **0.383** ±0.003 | 0.394 ±0.002 | 0.465 ±0.003 | 0.540 ±0.000 | 0.877 ±0.000 | 0.893 ±0.000 | 1.459 ±0.000 |
| | 96 | **0.385** ±0.003 | 0.400 ±0.004 | 0.411 ±0.002 | 0.464 ±0.002 | 0.539 ±0.000 | 0.860 ±0.000 | 0.859 ±0.000 | 1.444 ±0.000 |
| Electricity | 24 | **0.165** ±0.001 | 0.187 ±0.003 | 0.170 ±0.001 | 0.200 ±0.001 | 0.222 ±0.000 | 1.039 ±0.000 | 1.000 ±0.000 | 1.707 ±0.000 |
| | 48 | **0.188** ±0.003 | 0.203 ±0.008 | 0.207 ±0.003 | 0.218 ±0.000 | 0.238 ±0.000 | 1.014 ±0.000 | 0.987 ±0.000 | 1.729 ±0.000 |
| | 72 | **0.209** ±0.004 | 0.230 ±0.001 | 0.253 ±0.004 | 0.234 ±0.007 | 0.264 ±0.000 | 1.023 ±0.000 | 0.993 ±0.000 | 1.759 ±0.000 |
| | 96 | **0.211** ±0.001 | 0.230 ±0.014 | 0.316 ±0.002 | 0.237 ±0.001 | 0.264 ±0.000 | 1.013 ±0.000 | 0.971 ±0.000 | 1.747 ±0.000 |
| Solar | 24 | **0.446** ±0.002 | 0.472 ±0.003 | 0.473 ±0.001 | 0.612 ±0.006 | 0.828 ±0.000 | 0.999 ±0.000 | 1.001 ±0.000 | 1.869 ±0.000 |
| | 48 | **0.546** ±0.003 | 0.603 ±0.004 | 0.574 ±0.001 | 0.717 ±0.001 | 0.928 ±0.000 | 0.968 ±0.000 | 1.007 ±0.000 | 1.872 ±0.000 |
| | 72 | **0.666** ±0.003 | **0.667** ±0.004 | 0.698 ±0.002 | 0.766 ±0.006 | 0.978 ±0.000 | 0.974 ±0.000 | 1.002 ±0.000 | 1.855 ±0.000 |
| | 96 | **0.713** ±0.004 | 0.739 ±0.009 | 0.730 ±0.005 | 0.827 ±0.001 | 1.004 ±0.000 | 0.974 ±0.000 | 0.997 ±0.000 | 1.874 ±0.000 |

Table 2: Comparison of different denoising baselines to our forecast-corrupt-denoise approach with GPs when treating Autoformer forecasting model. We find that our approach consistently outperforms the other denoising approaches. Results are reported as average and standard error of **MSE**. A lower **MSE** indicates a better forecasting model.

| Dataset | Horizon | AutoDG(Ours) | Autoformer | AutoDI | AutoDWC | AutoRB | AutoDT |
|---|---|---|---|---|---|---|---|
| Traffic | 24 | **0.392** ±0.006 | 0.412 ±0.006 | 0.405 ±0.003 | 0.400 ±0.005 | 0.447 ±0.006 | 0.430 ±0.015 |
| | 48 | **0.387** ±0.001 | 0.422 ±0.007 | 0.416 ±0.007 | 0.417 ±0.009 | 0.450 ±0.005 | 0.410 ±0.005 |
| | 72 | **0.380** ±0.001 | **0.383** ±0.002 | 0.394 ±0.002 | 0.398 ±0.003 | 0.430 ±0.004 | 0.404 ±0.006 |
| | 96 | **0.385** ±0.003 | 0.400 ±0.004 | 0.411 ±0.002 | 0.405 ±0.001 | 0.413 ±0.002 | 0.422 ±0.002 |
| Electricity | 24 | **0.165** ±0.001 | 0.187 ±0.003 | 0.170 ±0.001 | 0.174 ±0.00 | 0.260 ±0.001 | 0.170 ±0.007 |
| | 48 | **0.188** ±0.003 | 0.203 ±0.008 | 0.207 ±0.003 | 0.219 ±0.002 | 0.222 ±0.002 | 0.200 ±0.002 |
| | 72 | **0.209** ±0.004 | 0.230 ±0.001 | 0.253 ±0.004 | 0.218 ±0.010 | 0.234 ±0.022 | 0.212 ±0.002 |
| | 96 | **0.211** ±0.001 | 0.230 ±0.014 | 0.316 ±0.002 | 0.226 ±0.008 | 0.296 ±0.011 | 0.218 ±0.004 |
| Solar | 24 | **0.446** ±0.002 | 0.472 ±0.003 | 0.473 ±0.006 | **0.449** ±0.003 | 0.527 ±0.006 | 0.457 ±0.004 |
| | 48 | **0.546** ±0.005 | 0.603 ±0.003 | 0.574 ±0.001 | 0.605 ±0.005 | 0.595 ±0.005 | 0.598 ±0.003 |
| | 72 | **0.666** ±0.003 | **0.667** ±0.004 | 0.698 ±0.002 | 0.690 ±0.010 | 0.718 ±0.002 | 0.670 ±0.006 |
| | 96 | **0.713** ±0.004 | 0.739 ±0.009 | 0.730 ±0.005 | 0.732 ±0.006 | 0.753 ±0.007 | 0.733 ±0.006 |

**Solar Energy** [3]: A univariate dataset about solar power that could be obtained across different locations in America, collected on an hourly interval.

**Electricity** [4]: A univariate dataset listing the electricity consumption of 370 customers, aggregated on an hourly level.

From each dataset, we roughly use 40,000 samples, where each sample contains given observations $X$ of $\kappa = 192$ time steps, from which (using multiple horizon forecasting) we predict the next $\tau \in \{24, 48, 72, 96\}$ future time steps. After Z-score normalization, we partition 40,000 samples of each dataset into three parts, 80% for training, 10% for validation, and 10% for performance evaluation.

**Evaluation metrics.** We evaluate our model and other alternatives using mean squared error $\text{MSE} = \frac{1}{n}(\sum_{t=1}^{n}(\boldsymbol{y}_t - \hat{\boldsymbol{y}}_t)^2)$, where $n$ denotes the length of the predicted time series. We also study the mean absolute error $\text{MAE} = \frac{1}{n}(\sum_{t=1}^{n}|\boldsymbol{y}_t - \hat{\boldsymbol{y}}_t|)$, where we obtain the same findings (omitted from this manuscript, but available in Appendix A).

## 3.2 TREATED TIME SERIES FORECASTING MODELS AND BASELINES

Since our corruption-resilient approach can be used to treat any forecasting model, we study the benefit for the following state-of-the-art time series forecasting models. The number of layers is tuned with Optuna.

**Autoformer Wu et al. (2021):** A multi-layer Autoformer model with auto-correlation

**Informer Zhou et al. (2021):** A multi-layer informer with ProbAttention.

We conduct a comparative analysis focusing on the following treatments applied to the Autoformer and Informer forecasting model:

**Auto(Info)DG (our proposed model):** Our proposed forecast-corrupt-denoise framework with Gaussian-Process-based corruption model as described in Section 2.

**Auto(Info)DI (denoise scaled Isotropic noise):** the same forecast-corrupt-denoise model, albeit using scaled isotropic Gaussians as a corruption model (instead of the GP).

**Auto(In)former (only forecasting):** the original untreated forecasting method.

We also compare against the follwoing baselines:

**ARIMA Hyndman & Khandakar (2008):** An autoregressive integrated moving average.

**CMGP Chakrabarty et al. (2021):** Model calibration using Bayesian Optimization and GPs.

**DeepAR Salinas et al. (2020):** An Autoregressive probabilistic time series forecasting model that uses LSTMs to estimate the parameters of a Gaussian distribution.

**DLinear Zeng et al. (2022):** A forecasting model that uses multi-layer perceptron to make predictions.

**Nbeats Oreshkin et al. (2020):** A neural approach for interpreting trend, seasonality, and residuals.

In an ablation study, we additionally compare against the following canonical denoising and boosting approaches for the Autoformer model. We obtain very similar results for the Informer model, omitted here but provided in Appendix A.

**AutoDWC (denoise without corruption):** a forecast-denoise model, where the denoising acts directly on the predictions (no corruption).

**AutoRB (residual-boosted):** two forecasting models, where the second is trained on minimizing the error residuals between the predictions and the ground-truth.

**AutoDT (denoise only during Training):** the forecasting model is trained to denoise with GP corruption, but the corruption is not used at test time.

---

[3] Solar energy https://www.nrel.gov/grid/assets/downloads/al-pv-2006.zip

[4] Electricity https://archive.ics.uci.edu/ml/machine-learning-databases/00321/LD2011_2014.txt.zip

Table 3: Overall results of the quantitative evaluation of corruption-resilient and baseline forecasting models in terms of average and standard error of **MSE**. We compare the forecasting models on all three datasets with different number of forecasting steps. A lower **MSE** indicates a better model. In all cases our forecast-corrupt-denoise approach with GPs improves performance of the original forecasting model (Inoformer) and isotropic Gaussian noise model (InfoDI). (Note that to provide a fair comparison, all the baseline models considered in this study were trained and evaluated under the *same* experimental setup as our proposed model. Consequently, the reported results may differ from those originally reported in the respective baseline papers. We provide the baseline models implementation in our online repository).

| Dataset | Horizon | **InfoDG(Ours)** | Informer | InfoDI | NBeats | DLinear | DeepAR | CMGP | ARIMA |
|---|---|---|---|---|---|---|---|---|---|
| Traffic | 24 | **0.398** ±0.006 | 0.421 ±0.006 | 0.415 ±0.003 | 0.475 ±0.008 | 0.553 ±0.000 | 0.888 ±0.000 | 0.824 ±0.000 | 1.436 ±0.000 |
| | 48 | **0.399** ±0.001 | 0.434 ±0.004 | 0.395 ±0.001 | 0.462 ±0.012 | 0.547 ±0.000 | 0.944 ±0.000 | 0.828 ±0.000 | 1.444 ±0.000 |
| | 72 | **0.380** ±0.001 | 0.436 ±0.001 | 0.395 ±0.002 | 0.465 ±0.002 | 0.540 ±0.000 | 0.877 ±0.000 | 0.893 ±0.000 | 1.459 ±0.000 |
| | 96 | **0.397** ±0.003 | **0.402** ±0.003 | **0.402** ±0.004 | 0.464 ±0.004 | 0.539 ±0.000 | 0.860 ±0.000 | 0.859 ±0.000 | 1.444 ±0.000 |
| Electricity | 24 | **0.193** ±0.001 | 0.222 ±0.001 | 0.212 ±0.003 | 0.200 ±0.001 | 0.222 ±0.000 | 1.039 ±0.000 | 1.000 ±0.000 | 1.707 ±0.000 |
| | 48 | **0.222** ±0.003 | 0.262 ±0.007 | 0.229 ±0.003 | 0.218 ±0.003 | 0.238 ±0.000 | 1.014 ±0.000 | 0.987 ±0.000 | 1.729 ±0.000 |
| | 72 | **0.238** ±0.001 | 0.280 ±0.004 | 0.253 ±0.004 | **0.234** ±0.007 | 0.264 ±0.000 | 1.023 ±0.000 | 0.993 ±0.000 | 1.759 ±0.000 |
| | 96 | 0.242 ±0.001 | 0.289 ±0.002 | 0.275 ±0.014 | **0.237** ±0.001 | 0.264 ±0.000 | 1.013 ±0.000 | 1.130 ±0.000 | 1.747 ±0.000 |
| Solar | 24 | **0.455** ±0.009 | 0.524 ±0.003 | **0.465** ±0.006 | 0.612 ±0.006 | 0.828 ±0.000 | 0.999 ±0.000 | 0.971 ±0.000 | 1.869 ±0.000 |
| | 48 | **0.556** ±0.005 | 0.629 ±0.003 | 0.570 ±0.005 | 0.717 ±0.001 | 0.928 ±0.000 | 0.968 ±0.000 | 1.007 ±0.000 | 1.872 ±0.000 |
| | 72 | **0.643** ±0.003 | 0.729 ±0.023 | 0.707 ±0.002 | 0.766 ±0.006 | 0.978 ±0.000 | 0.974 ±0.000 | 1.002 ±0.000 | 1.855 ±0.000 |
| | 96 | **0.708** ±0.004 | 0.770 ±0.004 | 0.766 ±0.009 | 0.827 ±0.005 | 1.004 ±0.000 | 0.974 ±0.000 | 0.997 ±0.000 | 1.874 ±0.000 |

## 3.3 MODEL TRAINING AND HYPER-PARAMETERS

All models are trained and evaluated three times using three different random seeds. We use Optuna Akiba et al. (2019) for hyper-parameter optimization. We tune the warm up steps of the optimization, model size (dimensionality of latent space) for all models, chosen from $\{16, 32\}$, and the number of layers of the forecasting model chosen from $\{1, 2\}$.

We use 8-head attention for all attention-based models. We model the GP using ApproximateGP of the GPyTorch package.[5] The batch size is set to 256. We use the Adam Kingma & Ba (2015) optimizer with $\beta_1 = 0.9$, $\beta_2 = 0.98$ and $\epsilon = 10^{-9}$, we change the learning rate following Vaswani et al. (2017) with warm-up steps chosen from $\{1000, 8000\}$. All models are trained on a single NVIDIA A40 GPU with 45GB of memory. We train our forecasting and denoising model with total number of 50 epochs. Training one epoch of our end-to-end model roughly takes about 25 seconds.

## 3.4 RESULTS AND DISCUSSION

Table 1 and 3 summarize the evaluation results of the treatments of the Autoformer and Informer forecasting models along with other baselines on the three datasets. Results are reported as average and standard errors in terms of MSE. All forecasting models are evaluated on their ability to predict for the next 24, 48, 72 and 96 future time steps. When treating the Autoformer and Informer

---

[5]https://gpytorch.ai/

models, our proposed GP-based corruption-resilient forecasting model predominately outperforms the Auto(Info)DI and the initial forecasting models. This funding is consistent across all data sets.

Next we conduct an ablation study by comparing our approach with several canonical denoising and boosting approaches. Table 2 presents an overview of the additional treatments applied to the Autoformer model. Compared to using a corruption/denoising approach during training (AutoDT), our model consistently performs better, supporting our claims. When attempting to denoise predictions directly without applying corruption, AutoDWC does not consistently yield improvements over the untreated forecasting model Autoformer. This highlights the significance of our GP corruption model in enhancing the resilience and accuracy of the forecasts. This inconsistency in improving the untreated forecasting model is apparent in the AutoDI model as well. This reinforces our hypothesis that denoising isotropic noise does not provide benefits for time series data. This applies when dealing with time series forecasting model with inaccurate predictions that do not exhibit jitters.

The success of our predict-corrupt-denoise model in comparison to traditional denoising models lies in the division of responsibilities arising from the corruption model. As a result the initial forecasting model focuses on predicting the broader patterns and trends, while a dedicated denoising forecaster addresses the finer details. This results in an overall more accurate forecasting model.

Please refer to Appendix A for other extensive ablation studies, examples of actual forecasts, and other supplementary materials.

## 4    SOCIETAL CONSEQUENCES

Time series forecasting offers societal benefits such as optimizing traffic lights, aligning energy grid capacities with solar power, and providing quantitative models for scientific understanding. However, like any foundational technology, it can also be misused for negative impacts, like aiding criminals or misinformation campaigns. By enhancing machine learning methods for more accurate forecasting, we aim to improve current practices, with the hope that the positive aspects will prevail.

## 5    CONCLUSION

In this paper, we study the multi-horizon time series forecasting problem and propose an end-to-end forecast-corrupt-denoise paradigm. In our proposed framework, we encourage the initial forecasting model to focus on accurately predicting the coarse-grained behavior, while the denoising model is responsible for predicting the fine-grained behavior. Both parts of the model communicate via a corruption model, which will blur the prediction in order to be denoised, and hence encourage a separation of concerns. This ultimately leads to more resilient forecasting methods. In addition to end-to-end training of the multi-component model, we utilize methods from variational techniques to guide the training of the denoising model via distribution matching.

Where most denoising methods leverage isotropic Gaussians, we hypothesize that a corruption process that resembles the true error modes of current forecasting models offers the most advantages. In line with Robinson et al. Robinson et al. (2018), we find that using a corruption model with temporal correlation, such as the Gaussian Process, is advantageous over uncorrelated noise models. Our experiments show that our proposed framework with Gaussian Process corruption is significantly outperforming the forecasting model without any denoising as well as denoising corruption of isotropic Gaussian noise. Additionally, we show that our forecast-corrupt-denoise framework predominately outperforms an approach that uses corruption/denoising only during training (AutoDT).

A strength of our approach is that it can be applied to any neural forecasting model. We demonstrate the effectiveness of our approach across three real-world datasets, different horizons, and several the-state-of-the-art forecasting models, including Autoformer and Informer (available in Appendix A). In all experiments, our proposed corruption-resilient forecasting approach with Gaussian Process corruption leads to significant improvements.

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

## A   APPENDIX

**Actual Forecasts:**  Please refer to Figure 3, 4, and 5 for the predicted forecasts of 72 future time steps for four treatments of Autoformer model including a) standalone forecasting b) AutoDI, c) AutoDWC, and d) AutoDG(ours) on Traffic, Electricity, and Solar datasets respectively .

**Convergence Plots:**  Please refer to Figure 6, 7, and 8 for the convergence plots of three treatments of the Autoformer model inclusing ours respectively.

**Main results of MAE metric:**  Please refer to Table 4 and 6 for main results of MAE metric respectively.

**Ablation results of other denoising/boosting baselines:**  Please refer to Table 5, 9, and 10 for the ablation results of Autoformer (MAE) and Informer (MSE and MAE) ablation studies on other denoising/boosting baselines respectively.

**Ablation results for higher number of layers (parameters) of the stand-alone forecasting models:** Please refer to Table 7 and 8 for the ablation study of higher number of layers (parameters) of stand-alone forecasting models respectively.

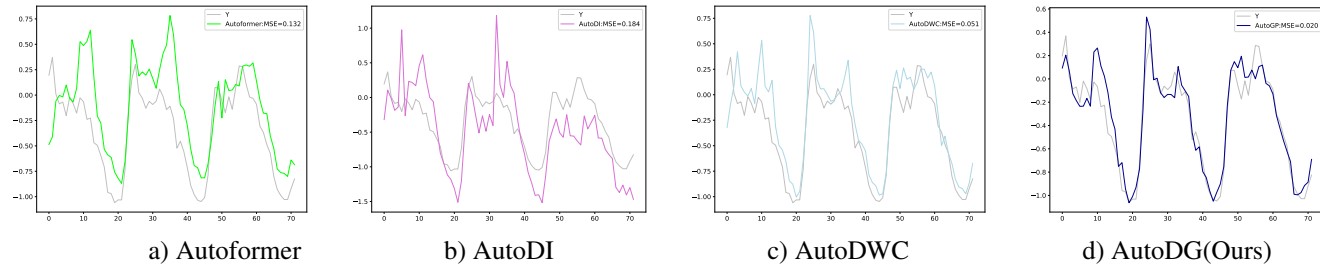

| a) Autoformer | b) AutoDI | c) AutoDWC | d) AutoDG(Ours) |

Figure 3: Example forecast of four treatments on the **Traffic** dataset for 72 future time steps using the Autoformer forecasting model. The values are plotted in Z-score normalized space. Standalone Autoformer model a) generally tracks the ground-truth, albeit fine-grained features are not accuratelt reproduced. Autoformer with isotropic corruption and denoising (AutoDI) b) yields a higher MSE with forecasts containing many jitters leading to inaccura te local behavior. Denoising without corruption (AutoDWC) c) yield a better MSE but fine-grained features including details and extreme values are not accurately reproduced. Our proposed Autoformer with Gaussian Process corruption and denoising (AutoDG) d) produces the most accurate forecasts by accurately predicting coarse-grained behavior of peaks and valleys, as well as fine-grained behavior such as smooth slopes, details and better extreme values prediction.

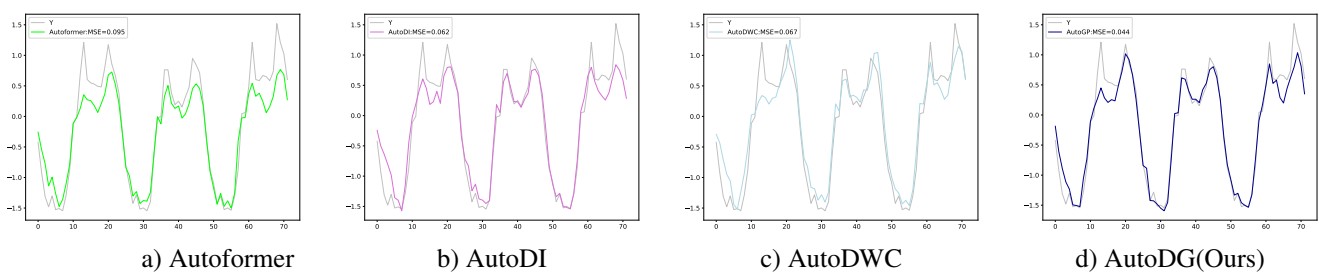

| a) Autoformer | b) AutoDI | c) AutoDWC | d) AutoDG(Ours) |

Figure 4: Example forecast of four treatments on the **Electricity** dataset for 72 future time steps using the Autoformer forecasting model. The values are plotted in Z-score normalized space. Standalone Autoformer model a) generally tracks the ground-truth, albeit fine-grained features are only roughly reproduced. Autoformer with isotropic corruption and denoising (AutoDI) b) yields a lower MSE with fine-grained features being more accurately predicted. Denoising without corruption (AutoDWC) c) yields a better MSE than Autoformer a), however fine-grained features are less accurately predicted than AutoDI. Our proposed Autoformer with Gaussian Process corruption and denoising (AutoDG) d) produces the most accurate forecasts by accurately predicting coarse-grained behavior of peaks and valleys, as well as fine-grained behavior such as smooth slopes, details and better extreme values prediction.

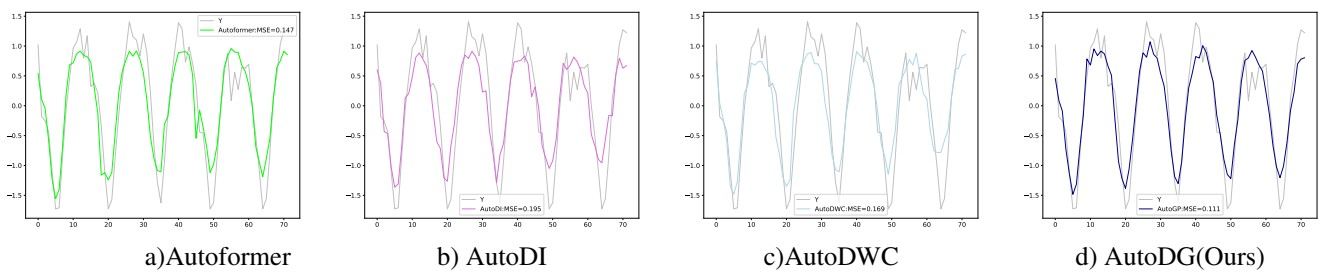

| a)Autoformer | b) AutoDI | c)AutoDWC | d) AutoDG(Ours) |

Figure 5: Example forecast of four treatments on the **Solar** dataset for 72 future time steps using the Autoformer forecasting model. The values are plotted in Z-score normalized space. Standalone Autoformer model a) generally tracks the ground-truth, albeit fine-grained features are only roughly reproduced. Autoformer with isotropic corruption and denoising b) yields a higher MSE with less accurate local behavior (e.g. prediction of extreme values). Denoising without corruption (AutoDWC) c) yields a higher MSE than Autoforme a) with fine-grained features being less accurately predicted. Our proposed Autoformer with Gaussian Process corruption and denoising (AutoDG) d) produces the most accurate forecasts by accurately predicting coarse-grained behavior of peaks and valleys, as well as fine-grained behavior such as smooth slopes, details and better extreme values prediction.

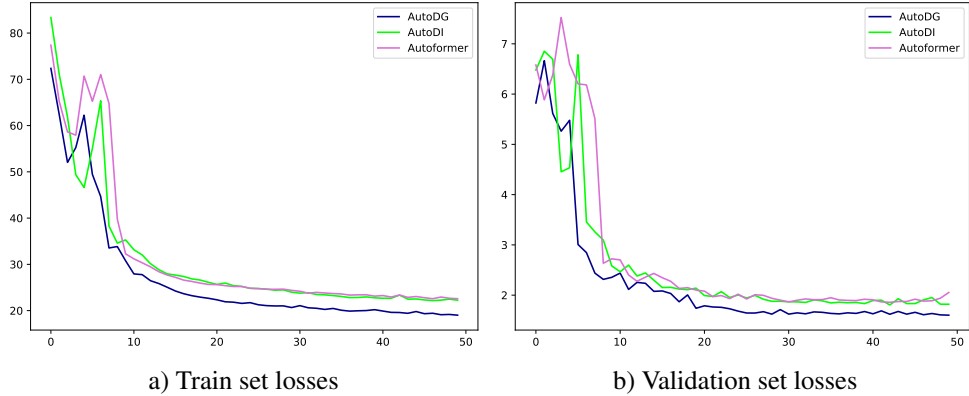

a) Train set losses                    b) Validation set losses

Figure 6: MSE loss illustration of Autformer, AutoDI, and AutoDG (Ours) of train a) and validation b) sets during training with 50 epochs on **Traffic** dataset when predicting for 72 future time steps. The plots demonstrate that our model offers a more favorable optimization space, showcasing superior convergence and absence of overfitting when compared to Autformer and AutoDI approaches.

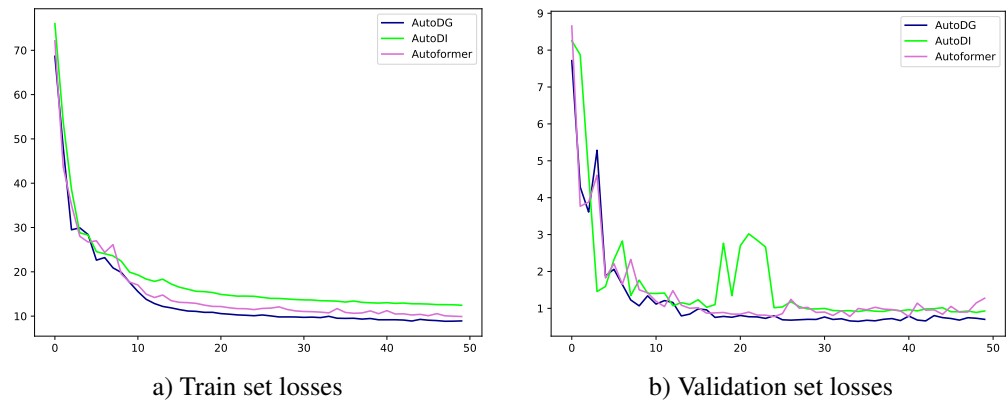

a) Train set losses                    b) Validation set losses

Figure 7: MSE loss illustration of Autformer, AutoDI, and AutoDG (Ours) of train a) and validation b) sets during training with 50 epochs on **Electricity** dataset when predicting for 72 future time steps. The plots demonstrate that our model offers a more favorable optimization space, showcasing superior convergence and absence of overfitting when compared to Autformer and AutoDI approaches.

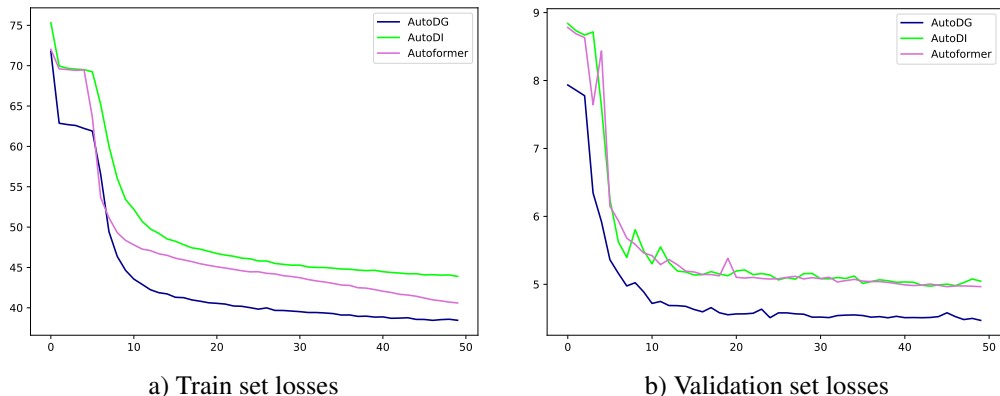

a) Train set losses                    b) Validation set losses

Figure 8: MSE loss illustration of Autformer, AutoDI, and AutoDG (Ours) of train a) and validation b) sets during training with 50 epochs on **Solar** dataset when predicting for 72 future time steps. The plots demonstrate that our model offers a more favorable optimization space, showcasing superior convergence and absence of overfitting when compared to Autformer and AutoDI approaches.

Table 4: Overall results of the quantitative evaluation of corruption-resilient forecasting models in terms of average and standard error of **MAE**. We compare the forecasting models on all three datasets with different number of forecasting steps. A lower **MAE** indicates a better model. In all cases our predict-corrupt-denoise approach with GPs improves performance of the original forecasting model (Autoformer). It is significantly better than isotropic Gaussian noise model (AutoDI). (Note that to provide a fair comparison, all the baseline models considered in this study were trained and evaluated under the *same* experimental setup as our proposed model. Consequently, the reported results may differ from those originally reported in the respective baseline papers. We provide the baseline models in our online repository).

| Dataset | Horizon | AutoDG(Ours) | Autoformer | AutoDI | NBeats | DLinear | DeepAR | CMGP | ARIMA |
|---|---|---|---|---|---|---|---|---|---|
| Traffic | 24 | **0.333** ±0.010 | **0.334** ±0.007 | **0.340** ±0.007 | 0.384 ±0.001 | 0.447 ±0.000 | 0.652 ±0.000 | 0.645 ±0.000 | 0.770 ±0.000 |
| | 48 | **0.328** ±0.001 | 0.368 ±0.006 | 0.343 ±0.006 | 0.408 ±0.003 | 0.462 ±0.000 | 0.650 ±0.000 | 0.642 ±0.000 | 0.776 ±0.000 |
| | 72 | **0.358** ±0.013 | **0.356** ±0.003 | **0.356** ±0.005 | 0.413 ±0.004 | 0.466 ±0.000 | 0.636 ±0.000 | 0.648 ±0.000 | 0.782 ±0.000 |
| | 96 | **0.333** ±0.000 | 0.359 ±0.004 | 0.366 ±0.004 | 0.414 ±0.002 | 0.471 ±0.000 | 0.632 ±0.000 | 0.647 ±0.000 | 0.773 ±0.000 |
| Electricity | 24 | **0.249** ±0.001 | 0.265 ±0.003 | 0.258 ±0.001 | 0.294 ±0.004 | 0.299 ±0.000 | 0.862 ±0.000 | 0.840 ±0.000 | 0.959 ±0.000 |
| | 48 | **0.275** ±0.003 | 0.292 ±0.007 | 0.301 ±0.003 | 0.310 ±0.007 | 0.308 ±0.000 | 0.853 ±0.000 | 0.839 ±0.000 | 0.971 ±0.000 |
| | 72 | **0.303** ±0.004 | **0.297** ±0.006 | **0.303** ±0.004 | 0.322 ±0.007 | 0.323 ±0.000 | 0.856 ±0.000 | 0.836 ±0.000 | 0.987 ±0.000 |
| | 96 | **0.304** ±0.001 | 0.372 ±0.010 | 0.325 ±0.002 | 0.324 ±0.002 | 0.329 ±0.000 | 0.850 ±0.000 | 0.832 ±0.000 | 0.983 ±0.000 |
| Solar | 24 | **0.548** ±0.009 | 0.603 ±0.002 | 0.574 ±0.008 | 0.632 ±0.001 | 0.801 ±0.000 | 0.885 ±0.000 | 0.885 ±0.000 | 1.100 ±0.000 |
| | 48 | **0.612** ±0.003 | 0.656 ±0.003 | 0.638 ±0.003 | 0.710 ±0.001 | 0.864 ±0.000 | 0.865 ±0.000 | 0.888 ±0.000 | 1.102 ±0.000 |
| | 72 | **0.702** ±0.0001 | 0.729 ±0.017 | **0.702** ±0.001 | 0.744 ±0.001 | 0.894 ±0.000 | 0.873 ±0.000 | 0.885 ±0.000 | 1.106 ±0.000 |
| | 96 | **0.725** ±0.000 | 0.754 ±0.009 | 0.747 ±0.005 | 0.781 ±0.002 | 0.911 ±0.000 | 0.866 ±0.000 | 0.882 ±0.000 | 1.097 ±0.000 |

Table 5: Comparison of different denoising baselines to our forecast-corrupt-denoise approach with GPs when treating Autoformer forecasting model. We find that our approach consistently outperforms the other denoising approaches. Results are reported as average and standard error of **MAE**. A lower **MAE** indicates a better forecasting model.

| Dataset | Horizon | AutoDG(Ours) | Autoformer | AutoDI | AutoDWC | AutoRB | AutoDT |
|---|---|---|---|---|---|---|---|
| Traffic | 24 | **0.333**±0.010 | **0.334**±0.007 | **0.340**±0.007 | 0.345±0.009 | 0.391±0.005 | 0.349±0.005 |
| | 48 | **0.328**±0.001 | 0.368±0.006 | 0.343±0.006 | 0.351±0.005 | 0.359±0.002 | 0.361±0.016 |
| | 72 | **0.358**±0.013 | **0.356**±0.003 | **0.356**±0.005 | 0.361±0.002 | 0.383±0.005 | 0.379±0.001 |
| | 96 | **0.304**±0.000 | 0.325±0.004 | 0.372±0.004 | 0.362±0.004 | 0.368±0.005 | 0.379±0.005 |
| Electricity | 24 | **0.249**±0.001 | 0.265±0.003 | 0.258±0.001 | 0.272±0.001 | 0.380±0.001 | 0.263±0.007 |
| | 48 | **0.275**±0.003 | 0.292±0.007 | 0.301±0.003 | 0.306±0.002 | 0.311±0.002 | 0.288±0.003 |
| | 72 | **0.303**±0.004 | **0.297**±0.006 | **0.303**±0.004 | **0.295**±0.009 | 0.330±0.021 | **0.305**±0.003 |
| | 96 | **0.304**±0.001 | 0.372±0.010 | 0.325±0.002 | 0.324±0.005 | 0.386±0.005 | 0.318±0.005 |
| Solar | 24 | **0.548**±0.009 | 0.603±0.002 | 0.574±0.008 | 0.549±0.008 | 0.601±0.005 | 0.598±0.006 |
| | 48 | **0.612**±0.003 | 0.656±0.003 | 0.638±0.003 | 0.656±0.002 | 0.645±0.003 | 0.655±0.003 |
| | 72 | **0.702**±0.001 | 0.729±0.017 | **0.702**±0.001 | 0.724±0.008 | 0.720±0.010 | 0.709±0.004 |
| | 96 | **0.725**±0.000 | 0.754±0.009 | 0.747±0.005 | 0.746±0.006 | 0.738±0.007 | 0.745±0.006 |

Table 6: Overall results of the quantitative evaluation of corruption-resilient forecasting models in terms of average and standard error of **MAE**. We compare the forecasting models on all three datasets with different number of forecasting steps. A lower **MAE** indicates a better model. In all cases our predict-corrupt-denoise approach with GPs improves performance of the original forecasting model (Informer). It is significantly better than isotropic Gaussian noise model (InfoDI). (Note that to provide a fair comparison, all the baseline models considered in this study were trained and evaluated under the *same* experimental setup as our proposed model. Consequently, the reported results may differ from those originally reported in the respective baseline papers. We provide the baseline models in our online repository).

| Dataset | Horizon | InfoDG(Ours) | Informer | InfoDI | NBeats | DLinear | DeepAR | CMGP | ARIMA |
|---|---|---|---|---|---|---|---|---|---|
| Traffic | 24 | 0.355 | **0.329** | 0.342 | 0.384 | 0.447 | 0.652 | 0.645 | 0.770 |
| | | ±0.007 | ±0.006 | ±0.004 | ±0.001 | ±0.000 | ±0.000 | ±0.000 | ±0.000 |
| | 48 | **0.350** | **0.354** | **0.362** | 0.408 | 0.462 | 0.650 | 0.642 | 0.776 |
| | | ±0.001 | ±0.012 | ±0.005 | ±0.003 | ±0.000 | ±0.000 | ±0.000 | ±0.000 |
| | 72 | **0.345** | 0.377 | 0.353 | 0.413 | 0.466 | 0.636 | 0.648 | 0.782 |
| | | ±0.003 | ±0.002 | ±0.008 | ±0.004 | ±0.000 | ±0.000 | ±0.000 | ±0.000 |
| | 96 | **0.397** | **0.402** | **0.402** | **0.414** | 0.471 | 0.632 | 0.647 | 0.773 |
| | | ±0.004 | ±0.011 | ±0.005 | ±0.002 | ±0.000 | ±0.000 | ±0.000 | ±0.000 |
| Electricity | 24 | **0.290** | **0.300** | **0.298** | **0.294** | **0.299** | 0.862 | 0.840 | 0.959 |
| | | ±0.008 | ±0.005 | ±0.003 | ±0.004 | ±0.000 | ±0.000 | ±0.000 | ±0.000 |
| | 48 | **0.311** | 0.349 | 0.325 | 0.310 | 0.308 | 0.853 | 0.839 | 0.971 |
| | | ±0.002 | ±0.009 | ±0.002 | ±0.007 | ±0.000 | ±0.000 | ±0.000 | ±0.000 |
| | 72 | **0.345** | 0.377 | 0.353 | 0.322 | 0.323 | 0.856 | 0.836 | 0.987 |
| | | ±0.003 | ±0.002 | ±0.008 | ±0.002 | ±0.000 | ±0.000 | ±0.000 | ±0.000 |
| | 96 | **0.342** | 0.378 | 0.379 | 0.324 | 0.329 | 0.850 | 0.832 | 0.983 |
| | | ±0.004 | ±0.011 | ±0.005 | ±0.002 | ±0.000 | ±0.000 | ±0.000 | ±0.000 |
| Solar | 24 | **0.533** | 0.597 | 0.563 | 0.632 | 0.801 | 0.885 | 0.885 | 1.100 |
| | | ±0.005 | ±0.002 | ±0.003 | ±0.001 | ±0.000 | ±0.000 | ±0.000 | ±0.000 |
| | 48 | **0.624** | 0.681 | 0.635 | 0.710 | 0.864 | 0.865 | 0.888 | 1.102 |
| | | ±0.007 | ±0.013 | ±0.005 | ±0.001 | ±0.000 | ±0.000 | ±0.000 | ±0.000 |
| | 72 | **0.690** | 0.752 | 0.735 | 0.744 | 0.894 | 0.873 | 0.885 | 1.106 |
| | | ±0.013 | ±0.017 | ±0.019 | ±0.001 | ±0.000 | ±0.000 | ±0.000 | ±0.000 |
| | 96 | **0.727** | 0.772 | 0.764 | 0.781 | 0.911 | 0.866 | 0.882 | 1.097 |
| | | ±0.006 | ±0.012 | ±0.004 | ±0.002 | ±0.000 | ±0.000 | ±0.000 | ±0.000 |

Table 7: Comparison of our forecast-corrupt-denoise approach with GPs with standalone forecasting models with higher number of layers (parameters) denoted by † sign. Initially, the number of layers for our proposed model and other baselines are chosen from $\{1, 2\}$, however to show that the performance of our model is indeed stems from its mechanism, we included the results of stand-alone forecasting models with number of layers chosen from $\{3, 4\}$. Results are reported as average and standard error of **MSE**. A lower **MSE** indicates a better forecasting model.

| Dataset | Horizon | AutoDG(Ours) | Autoformer | Autoformer† | InfoDG(Ours) | Informer | Informer† |
|---|---|---|---|---|---|---|---|
| Traffic | 24 | 0.392 ±0.006 | 0.412 ±0.006 | **0.359** ±0.007 | **0.398** ±0.006 | 0.421 ±0.006 | 0.422 ±0.009 |
| | 48 | 0.387 ±0.001 | 0.422 ±0.007 | **0.383** ±0.001 | **0.399** ±0.001 | 0.434 ±0.001 | 0.486 ±0.010 |
| | 72 | **0.380** ±0.001 | **0.383** ±0.002 | 0.442 ±0.006 | **0.380** ±0.001 | 0.436 ±0.001 | 0.412 ±0.003 |
| | 96 | **0.385** ±0.003 | 0.400 ±0.004 | 0.416 ±0.001 | 0.397 ±0.003 | **0.402** ±0.003 | 0.408 ±0.005 |
| Electricity | 24 | **0.165** ±0.001 | 0.187 ±0.003 | 0.242 ±0.007 | **0.193** ±0.001 | 0.222 ±0.001 | 0.266 ±0.001 |
| | 48 | **0.188** ±0.003 | 0.203 ±0.008 | 0.232 ±0.005 | **0.222** ±0.002 | 0.262 ±0.002 | 0.293 ±0.002 |
| | 72 | **0.209** ±0.004 | 0.230 ±0.001 | 0.263 ±0.004 | **0.238** ±0.001 | 0.280 ±0.003 | 0.310 ±0.002 |
| | 96 | **0.211** ±0.001 | 0.230 ±0.014 | 0.224 ±0.004 | **0.242** ±0.001 | 0.289 ±0.002 | 0.327 ±0.003 |
| Solar | 24 | **0.446** ±0.002 | 0.472 ±0.003 | 0.524 ±0.001 | **0.455** ±0.009 | 0.524 ±0.003 | 0.498 ±0.001 |
| | 48 | **0.546** ±0.005 | 0.603 ±0.003 | 0.622 ±0.001 | **0.556** ±0.005 | 0.629 ±0.003 | 0.690 ±0.031 |
| | 72 | **0.666** ±0.003 | **0.667** ±0.004 | 0.701 ±0.004 | **0.643** ±0.003 | 0.729 ±0.023 | 0.716 ±0.024 |
| | 96 | **0.713** ±0.004 | 0.739 ±0.009 | 0.744 ±0.002 | **0.708** ±0.004 | 0.770 ±0.004 | 0.738 ±0.002 |

Table 8: Comparison of our forecast-corrupt-denoise approach with GPs with standalone forecasting models with higher number of layers (parameters) denoted by † sign. Initially, the number of layers for our proposed model and other baselines are chosen from $\{1, 2\}$, however to show that the performance of our model is indeed stems from its mechanism, we included the results of stand-alone forecasting models with number of layers chosen from $\{3, 4\}$. Results are reported as average and standard error of **MAE**. A lower **MAE** indicates a better forecasting model.

| Dataset | Horizon | AutoDG(Ours) | Autoformer | Autoformer† | InfoDG(Ours) | Informer | Informer† |
|---|---|---|---|---|---|---|---|
| Traffic | 24 | **0.333** ±0.010 | **0.334** ±0.007 | **0.332** ±0.002 | 0.355 ±0.007 | **0.329** ±0.006 | 0.382 ±0.003 |
| | 48 | **0.328** ±0.001 | 0.368 ±0.002 | 0.336 ±0.001 | **0.345** ±0.001 | 0.377 ±0.013 | 0.402 ±0.005 |
| | 72 | **0.358** ±0.013 | **0.356** ±0.003 | 0.357 ±0.001 | **0.345** ±0.003 | 0.377 ±0.010 | 0.382 ±0.011 |
| | 96 | **0.385** ±0.003 | 0.400 ±0.004 | 0.370 ±0.001 | 0.397 ±0.003 | **0.402** ±0.002 | 0.413 ±0.003 |
| Electricity | 24 | **0.249** ±0.001 | 0.265 ±0.003 | 0.303 ±0.003 | **0.290** ±0.003 | **0.300** ±0.006 | 0.332 ±0.002 |
| | 48 | **0.275** ±0.001 | 0.292 ±0.007 | 0.317 ±0.002 | **0.311** ±0.002 | 0.349 ±0.009 | 0.377 ±0.002 |
| | 72 | **0.303** ±0.004 | **0.297** ±0.007 | 0.351 ±0.002 | **0.336** ±0.003 | 0.371 ±0.002 | 0.384 ±0.002 |
| | 96 | **0.304** ±0.001 | 0.372 ±0.010 | 0.317 ±0.001 | **0.342** ±0.004 | 0.378 ±0.001 | 0.411 ±0.003 |
| Solar | 24 | **0.548** ±0.009 | 0.603 ±0.002 | 0.608 ±0.001 | **0.533** ±0.005 | 0.597 ±0.002 | 0.575 ±0.000 |
| | 48 | **0.612** ±0.003 | 0.656 ±0.003 | 0.672 ±0.004 | **0.624** ±0.007 | 0.681 ±0.013 | 0.745 ±0.020 |
| | 72 | **0.702** ±0.001 | 0.729 ±0.017 | 0.707 ±0.001 | **0.690** ±0.013 | 0.752 ±0.017 | 0.762 ±0.016 |
| | 96 | **0.725** ±0.000 | 0.754 ±0.009 | 0.737 ±0.002 | **0.727** ±0.006 | 0.772 ±0.012 | 0.785 ±0.010 |

Table 9: Comparison of different denoising baselines to our forecast-corrupt-denoise approach with GPs when treating Informer forecasting model. We find that our approach consistently outperforms the other denoising approaches. Results are reported as average and standard error of **MSE**. A lower **MSE** indicates a better forecasting model.

| Dataset | Horizon | InfoDG(Ours) | Informer | InfoDI | InfoDWC | InfoRB | InfoDT |
|---------|---------|--------------|----------|--------|---------|--------|--------|
| Traffic | 24 | **0.398**±0.005 | 0.421±0.005 | 0.415±0.002 | 0.406±0.002 | 0.435±0.005 | 0.473±0.003 |
| | 48 | **0.399**±0.004 | 0.434±0.014 | 0.395±0.007 | 0.392±0.003 | **0.395** ±0.011 | 0.421±0.014 |
| | 72 | **0.380**±0.001 | 0.436±0.015 | 0.395±0.001 | 0.392±0.001 | 0.407±0.009 | 0.421±0.011 |
| | 96 | **0.397**±0.003 | 0.402±0.002 | **0.402**±0.006 | **0.394**±0.003 | 0.412±0.007 | 0.414±0.015 |
| Electricity | 24 | **0.193**±0.003 | 0.222±0.006 | 0.212±0.001 | 0.204±0.005 | 0.225±0.009 | 0.230±0.007 |
| | 48 | **0.222**±0.003 | 0.262±0.013 | 0.229±0.003 | 0.241±0.007 | 0.261±0.014 | 0.256±0.004 |
| | 72 | **0.238**±0.001 | 0.280±0.006 | 0.253±0.006 | 0.263±0.013 | 0.262±0.008 | 0.268±0.008 |
| | 96 | **0.242**±0.004 | 0.289±0.011 | 0.275±0.005 | 0.279±0.006 | 0.283±0.001 | 0.275±0.007 |
| Solar | 24 | **0.455**±0.007 | 0.524±0.002 | 0.465±0.006 | 0.457±0.006 | 0.498±0.010 | 0.512±0.012 |
| | 48 | **0.556**±0.011 | 0.629±0.021 | 0.570±0.007 | 0.590±0.016 | 0.623±0.016 | 0.629±0.023 |
| | 72 | **0.643**±0.022 | 0.729±0.024 | 0.707±0.026 | 0.708±0.014 | 0.748±0.010 | 0.726±0.006 |
| | 96 | **0.708**±0.010 | 0.770±0.017 | 0.766±0.006 | 0.739±0.010 | 0.781±0.017 | 0.777±0.000 |

Table 10: Comparison of different denoising baselines to our forecast-corrupt-denoise approach with GPs when treating Informer forecasting model. We find that our approach consistently outperforms the other denoising approaches. Results are reported as average and standard error of **MAE**. A lower **MAE** indicates a better forecasting model.

| Dataset | Horizon | InfoDG(Ours) | Informer | InfoDI | InfoDWC | InfoRB | InfoDT |
|---------|---------|--------------|----------|--------|---------|--------|--------|
| Traffic | 24 | 0.355±0.007 | **0.329**±0.006 | 0.342±0.004 | 0.337±0.003 | 0.331±0.003 | 0.379±0.003 |
| | 48 | **0.345**±0.001 | 0.377±0.013 | 0.353±0.005 | 0.348±0.003 | 0.353±0.009 | 0.375±0.006 |
| | 72 | **0.345**±0.003 | 0.377±0.010 | 0.353±0.004 | 0.348±0.001 | 0.379±0.005 | 0.361±0.006 |
| | 96 | **0.350**±0.004 | **0.354**±0.006 | **0.362**±0.007 | **0.348**±0.008 | 0.379±0.005 | **0.361**±0.011 |
| Electricity | 24 | **0.290**±0.008 | **0.300**±0.005 | **0.298**±0.003 | **0.295**±0.004 | **0.302**±0.009 | 0.318±0.003 |
| | 48 | **0.311**±0.002 | 0.349±0.009 | 0.325±0.002 | 0.333±0.004 | 0.343±0.009 | 0.343±0.005 |
| | 72 | **0.336**±0.003 | 0.371±0.002 | 0.359±0.008 | 0.362±0.008 | 0.359±0.006 | 0.367±0.008 |
| | 96 | **0.342**±0.004 | 0.378±0.0011 | 0.379±0.005 | 0.384±0.006 | 0.375±0.001 | 0.370±0.007 |
| Solar | 24 | **0.533**±0.005 | 0.597±0.002 | 0.563±0.003 | 0.551±0.001 | 0.573±0.008 | 0.596±0.007 |
| | 48 | **0.624**±0.007 | 0.681±0.013 | 0.635±0.005 | 0.649±0.011 | 0.675±0.012 | 0.681±0.012 |
| | 72 | **0.690**±0.013 | 0.752±0.017 | 0.735±0.019 | 0.736±0.011 | 0.763±0.020 | 0.735±0.004 |
| | 96 | **0.727**±0.006 | 0.772±0.012 | 0.764±0.004 | 0.753±0.005 | 0.777±0.014 | 0.766±0.002 |

