# OpenReview forum: "Gaussian Process-Based Corruption-resilience Forecasting Models"
_ICLR.cc/2024/Conference — Submitted to ICLR 2024_

### Official Review · Reviewer_nq8v · 2023-10-22

**Soundness:** 2 fair
**Presentation:** 1 poor
**Contribution:** 1 poor
**Rating:** 3
**Confidence:** 4

**Summary:**

The authors claim to have introduced a joint forecast-corrupt-denoise model. The output of the baseline forecasting model, for instance, a machine learning (ML) algorithm, is corrupted by a noise function following a GP distribution. Once data have been corrupted, a denoising model is deployed to reverse the corruption process, while seeking to improve the initial forecast output. Both parameters of the forecast and GP models are jointly learned via the minimization of a compound (forecast + GP-ELBO) loss function.

The authors also claim that their framework provides better results than the baseline forecasting models. To prove this, they have considered several experimental setups.

**Strengths:**

Under Gaussian assumptions, the consideration of a GP-based corruption process may be seen as an interesting idea for dealing with non-i.i.d. noise. Based on [16], an adapted compound loss function is proposed in the (forecasting and GP) parameter estimation. Python codes are provided.

**Weaknesses:**

- In my opinion, the contributions in the paper are minor. The authors have only adapted a collection of well-known approaches related to forecasting, GP and denoising models to establish their joint forecast-corrupt-denoise framework.
- Contrary to the authors' claims, the numerical results are not convincing. In many cases, the best results are not properly highlighted or are unclear since only 3 random replicates have been considered. For instance, in Table 4 (Traffic 48), InfoDWC (and possibly InfoDI) provides a better result than InfoDG (the proposed method). I suggest considering more replicates when constructing the tables to obtain more consistent results.
- There is no theoretical evidence (certification) to explain why the joint model should perform better than the baseline algorithms.
- Mathematical formulas are not defined correctly and, in some cases, are inconsistent. For instance, formulas related to GPs.
- The paper seems a bit rushed.

I will refer to the part **Questions** for further details.

**Questions:**

- According to Section 2.5, the GP-based corruption model is trained considering the ground truth $Y$ (data to forecast). Does it mean that the proposed joint model can be used only when $Y$ is known? If so, I don't see the point of setting up the problem as a forecast one instead of including $Y$ in the training dataset. If $Y$ is unknown (forecast context), how can the GP parameters be tuned?
- Can the authors explain the need to include two metrics (MSE and MAE) in the experimental setups? Since both metrics seek to assess the quality of predictions, the results are redundant. Consequently, half of the tables can be omitted.
- In the experimental setups, only 3 random replicates have been considered. Can the authors confirm that results are indeed consistent (i.e. that similar results are obtained for another triple of random replicates)? If not, they should consider a (statistically rich) number of random replicates (e.g. 10, 20, 30...).
- Page 8, Section 3.3: can the authors give further details on the choices (e.g. $\beta_1 = 0.9$, $\beta = 0.98$, and batch size = 256) considered in the numerical implementations?
- On page 1, Section 1, the authors suggest that their approach may be adapted to deal with multivariate transient functions but that their focus was on the univariate case. Have they performed numerical examples involving multivariate functions? In any case, can they provide further details on the scalability (in terms of the input dimension) of the framework?

**Other minor remarks**
- To cite references in brackets, e.g. [n], to avoid confusion when referring to equations (n).
- Page 1, Section 1 (Time series forecasting task): the definitions of $X$ and $Y$ are not clear. $\kappa$ is defined as the number of time-series observations prior to $t_0$, i.e. it defines a set of time instants $\{t_{-\kappa}, \ldots, t_{-1}\}$. Then $X$ needs to be defined as $\{x_t; \gamma_t\}_{t = t_0 - t_{-\kappa}}^{t_{-1}}$. A similar reasoning must be done for $Y = \{x_t; \gamma_t\}_{t = t_0}^{t_0 + t_\tau}$.
- Page 2, related works: LSTM is not defined.
- Page 3, 2nd paragraph: "navive"
- Page 3, Section 2.1: $\tilde{X}| X \sim \mathcal{N}(\textbf{0}, \sigma^2  \textbf{I})$ ($\sigma^2$ is missing)
- Tables are not placed just before or after their citation. For instance, Tables 1, 2 and 3 are placed on pages 4, 6 and 7 (respectively) but they are cited on page 8.
- Page 4, Eq. (1): the authors have defined $c$ as a function (since it is considered as a GP) but it is treated as a (Gaussian) vector. They need to be consistent with the definitions and distinguish them correctly throughout the paper.
- Punctuation marks in the equations need to be double-checked throughout the paper.
- To refer to GP everywhere once the abbreviation is introduced.
- Page 7, Section 3.1: footnotes 2, 3 and 4 are not provided. If the authors refer to references [2,3,4], they need to be cited properly.
- Reference 24: incompleted.

---

> ### Author Response · Authors · 2023-11-19
>
> Thank you for your comments. We updated the paper to meet your immediately actionable suggestions (omitted from the rebuttal). Please see below our responses to your questions:
>
> Questions:
>
> 1. Target variable $Y$ is only known at training time, at test time we are predicting the forecast $Y$, without assuming this knowledge.
>
> 2. It is common practice to include both MSE and MAE as evaluation metrics in regression tasks. Although, we only included the results for MAE in the Appendix section.
>
> 3. The choices of batch size and optimization parameters are based on our preliminary experiments.
>
> 4. Univariate forecasting is employed in this context because the datasets featured in the paper are exclusively univariate. In scenarios where the goal is to predict multiple target variables, the objective would be to optimize the MSE loss for multiple targets rather than a single one.

---

> > ### Comment · Reviewer_nq8v · 2023-11-20
> >
> > > 1. Target variable $Y$ is only known at training time, at test time we are predicting the forecast $Y$, without assuming this knowledge.
> >
> > According to the notation used in the paper, $\gamma$ (is this the target variable $Y$ at training time?) is the target function considering $x$ values up to $t_0$, and $Y$ is the target function after $t_0$ (see Fig. 2). In the definition of the compound loss function, the authors propose to minimize:
> > $$\mathcal{L} = L_{MSE}(\hat{Y} = Y | X) + \lambda  L_{ELBO}(Y_C = Y | Y_F).$$
> > Both objectives depend on $Y$, which is assumed to be unknown in forecasting tasks. This contradicts the authors' answer.
> >
> > Then, the authors need to double-check their notation to make things clearer or to justify the use of $Y$ when learning the covariance parameters (see my initial remark).
> >
> > > The choices of batch size and optimization parameters are based on our preliminary experiments.
> >
> > What kind of experiments? I believe it is necessary to give further details about their tuning to help potential readers interested in implementing their method.
> >
> > > Univariate forecasting is employed in this context because the datasets featured in the paper are exclusively univariate. In scenarios where the goal is to predict multiple target variables, the objective would be to optimize the MSE loss for multiple targets rather than a single one.
> >
> > The authors have pointed out on Page 1, Section 1, Time series forecasting task:
> > "The target variable can be multivariate with $\gamma_t \in \mathbb{R}^{d_y}$, although we focus on datasets with univariate target variables ($d_y = 1$)."
> > I was expecting to have further details about potential technical issues related to scalability when considering multivariate functions, i.e. $\gamma_t : \mathbb{R}^{d_y} \to \mathbb{R}$.

---

> ### Author Response · Authors · 2023-11-20
>
> > According to the notation used in the paper, $\gamma$ (is the target variable $Y$ at training time?) is the target function considering $x$ values up to $t_0$, and $Y$ is the target function after $t_0$ (see Fig.2). In the definition of the compound loss function, the authors propose to minimize:
>  $\mathcal{L} = L_{\mathtt{MSE}}(\hat{Y}=Y|X) + \lambda L_{\mathtt{ELBO}}(Y_C=Y|Y_F)$
>  Both objectives depend on $Y$ which is assumed to be unknown in forecasting tasks. This contradicts the authors' answer.
>
> We believe there is a misunderstanding here,  $\gamma$ is the target variable observed up to the cutoff point $t_0$, and $Y$ is the target variable that we are interested to forecast from $t_0$ to $t_0 + \tau$. Both are available during training but $Y$ is held-out during the test time. When referring to forecasting in SOTA _supervised_ (neural/no-neural) time series forecasting models, $Y$ is always known during training, in fact this is how the parameters of the models are optimized. Only traditional _un-supervised_ time series forecasting models including ARIMA do not rely on $Y$ during training, which is not the case if considering hyper-parameter tuning, therefore variable $Y$ can as well be known during training for unsupervised models.
>
> > What kind of experiments? I believe it is necessary to give further details about their tuning to help potential readers interested in implementing their method.
>
> We want to state that all hyper-parameters can further be fine-tuned for potential readers interested in implementing. However, the choices made was based on consistent better results for all datasets and all forecasting models, therefore opted as a part in hyper-parameter tuning.
>
> > The authors have pointed out on Page 1, Section 1, Time series forecasting task: "The target variable can be multivariate with
> $\gamma_t \in \mathbb{R}^ {d_y}$, although we focus on datasets with univariate target variables ( $d_y$ = 1
> )." I was expecting to have further details about potential technical issues related to scalability when considering multivariate functions, i.e. $\gamma_t : \mathbb{R} \rightarrow \mathbb{R}$.
>
> We want to further emphasize that all the variables are zero-mean normalized to prevent any scalability problem, additionally, further normalization techniques are applied when training models to enforce resiliency during training and test. Therefore, the task can be adapted for multi-variate forecasting by optimizing for multiple target variables rather than one.

---

> > ### Comment · Reviewer_nq8v · 2023-11-22
> >
> > I thank the authors for their reply. Based on my concerns and taking into account those of the other reviewers, I still think that the paper is below the bar. I have decided to increase my score to 3 (rejection, not good enough).

---

### Official Review · Reviewer_Niyb · 2023-10-23

**Soundness:** 3 good
**Presentation:** 3 good
**Contribution:** 2 fair
**Rating:** 5
**Confidence:** 3

**Summary:**

The authors propose a time-series forecasting module which can be applied to a wide-range of existing models. The main idea is denoising: the output of an already available forecasting model is corrupted by noise (the authors show the significance of correlated rather than i.i.d. noise) which is then passed through the same forecasting model (but with different parameters) again. The authors argue such an architecture "encourages the initial forecasting model to focus on modelling coarse-grained behavior, and a denoising model that corrects the fine-grained details". The proposed module is experimentally shown to improve the forecasting performance of a number of existing forecasting models.

**Strengths:**

+ An interesting idea combining the image denoising ideas with the time-series forecasting
+ Extensive experimental evaluation including the ablation study

**Weaknesses:**

- A somewhat confusing presentation of the proposed method (see the questions below)
- Lack of examples of the model forecasts apart from the cartoon in Fig. 1
- Minor grammatical errors and repetitions (e.g. almost the same sentence appears twice in Section 2.2.)

I have conflicting opinions about this paper. On the one hand, the experimental results look really good: the proposed denoising module noticeably improves the performance of the exiting models. On the other, I struggle to understand why it is the case, why adding noise improves the performance. In the image generation literature, denoising is often used as a tool for regularising the low-dimensional latent space which is used for sampling new images. However, it is clearly not the case in the context of time-series forecasting. I appreciate that the authors provided some intuition (e.g. see the quote in the Summary section of this review) but I would also appreciate further comments from the authors on this matter. I would be happy to increase my rating if the authors clarify some of these questions.

**Questions:**

- Why does the AutoDWC baseline (i.e. denoising without corruption) works worse than the noise corrupted model? Shouldn't adding independent (from the forecasting model prediction) noise make the task harder for the denoiser and thus deteriorate the performance?
- The isotropic noise baseline (AutoDI) performs similarly to the AutoDWC. Why do you think it is the case? Also how did you choose the variance of the isotropic noise?
- What is the input to the denoising model? Only the corrupted forecast, or the corrupted forecast and the historical time-series trajectory (from t_0-k to t_0) used to compute the forecast?
- Did you try different values of \lambda in Eq. (2)?

---

> ### Author Response · Authors · 2023-11-19
>
> Thank you for feedback and constructive comments. We updated the paper to meet your immediately actionable suggestions (omitted from the rebuttal). Please see below our responses to your comments:
>
> Weakness:
>
> 1. (& question 1) The success of our proposed model stems from the division of labors between the first forecasting stage (for coarse-grained) and the denoising stage (fine-grained predictions), where initial forecasts are blurred locally by utilizing a GP model. In between the parameters of the GP model are trained for best end-to-end performance.
>
> 2. To meet your request, we included figures with actual forecasts from our datasets in the paper and the provided link below:
>
> [Actual Forecasts](https://anonymous.4open.science/r/Corruption-resilient-Forecasting-Models-15E8/Actual-forecasts-including-AutoDWC.pdf)
>
> Questions:
>
> 2. The reason stems from the ineffectiveness of removing uncorrelated isotropic noise which is not the typical error mode in SOTA time series forecasting models. Therefore, the inability of AutoDI to outperform AutoDWC is underscored by such limitation. We adjusted the variance of the isotropic noise by fine-tuning the scaled factor $\sigma \in [0, 1]$.
>
> 3. The input to the denoising model is solely the corrupted (blurred) forecast, as our denoising model is designed to
> denoise the corrupted/blurred forecast to align with the ground-truth.
>
> 4. in our preliminary analysis setting different values for $\lambda$ did not affect the results drastically, however we will include more experiments of adjusting the variable $\lambda$ in Appendix of the camera-ready paper.

---

> ### Comment · Reviewer_Niyb · 2023-11-20
>
> Thank you for your reply and the additional figure with the model forecasts.
>
> I am still somewhat confused why adding independent noise helps in comparison to applying the same model twice (i.e. AutoDWC baseline) and for now I decided not to change my score.

---

> ### Author Response · Authors · 2023-11-20
>
> Thank you for your response.
>
> Let us further explain the intuition of our framework. By introducing the GP competent, we envision for a division of responsibilities, where the initial forecasting model predicts the overall (coarse-grained) behavior of the target variable, GP is integrated to introduce local blurring to the initial forecasts. In response, the denoising model focuses on removing the introduced blur by predicting the fine-grained details of the ground-truth.
>
> The inclusion of the GP component establishes a clear division of tasks between forecasting and denoising. While in the absence of the GP component, the distinction between coarse-grained and fine-grained details becomes less pronounced, potentially causing the denoiser to deprioritize the accurate prediction of fine-grained details. We would also like to emphasize that the parameters of the GP/corruption model are fine-tuned to obtain the best performance.
>
> We are also open to providing forecast plots of AutoDWC in comparison to our AutoDG(ours) to further illustrate why the division of tasks through GP integration helps to achieve enhanced performance.

---

> > ### Author Response · Authors · 2023-11-21
> >
> > We have included the actual forecasts of AutoDWC in comparison to our AutoDG model and two other treatments including Autoformer and AutoDI in the revised version of paper and the link below:
> >
> > [Actual forecasts including AutoDWC](https://anonymous.4open.science/r/Corruption-resilient-Forecasting-Models-15E8/Actual-forecasts-including-AutoDWC.pdf)

---

> > > ### Comment · Reviewer_Niyb · 2023-11-22
> > >
> > > Thank you for further clarifications and additional figure. I still can't say I have now a good intuition of the method, but it is not your fault, I guess I am just not familiar enough with such approaches. I appreciate the effort you put into the rebuttal discussions!
> > >
> > > To me it is still a borderline submission, so I keep my score as it is.

---

### Official Review · Reviewer_zFSw · 2023-10-29

**Soundness:** 3 good
**Presentation:** 2 fair
**Contribution:** 2 fair
**Rating:** 5
**Confidence:** 4

**Summary:**

This paper investigates the topic of noise corruption in modeling time-series data for forecasting. The authors propose that the noise/error in the time series data can be attributed to two sources, including a temporally correlated source, and an independent noise. From that the authors claim the current methods cannot well recover the underlying signals/true observations and thereby cannot carry out accurate forecast. To this end, the authors proposed a joint-corrupt-denoise model to capture the characteristics of signals from both sources identified. The framework is then tested on a wide range of datasets from which its efficacy is demonstrated.

**Strengths:**

1. The motivation of the problem is laid out very clearly, with an illustration well explaining the origin of the issue and the shortcomings of the current methods.
2. The proposed metholodgy is explained very clearly and a thorough and comprehensive numerical study is carried out to evaluate the method.

**Weaknesses:**

1. I have concerns that the proposed framework including the temporally-correlated term utilizing Gaussian process is entirely new. I think there might have been other works out there proposing a fairly similar idea called "calibration modeling". To elaborate a bit more here, the framework proposed carry some common components to a typical statistical calibration model as follows:
$y(t) = x(t) + \delta(t) + \epsilon(t),$
where $x(t)$ is the true signal, $\delta(t)$ is modeled by a Gaussian process, and $\epsilon(t)$ modeled by a Gaussian noise.
With an appropriate selection of the prior distributions for the parameters and hyperparameters, this can be tackled by a Bayesian approach. I will elaborate further in the Q section.
2. The presentation of the manuscript can be further improved. For example, in the abstract, I find the mentioning of "using more training data" and "image generation" not necessarily closely related to the point I think the authors were trying to emphasize. Additionally, in Figure 2, $X_1$ and $X_2$ are defined as covariates, which seem to conflict with the previos reference to $X$ as the observations.

**Questions:**

Let me further expand on the proposed framework itself. I believe the joint framework is quite novel and has clear potential in improving time series forecasting, as demonstrated by the authors very diligently. However, I do wonder whether the authors would be open to compare and evaluate the framework to the calibration model that I mentioned above.
My personal belief is they share some commonality between them in how they model the time-correlated noise/signal, but it seems the two objectives functions are still quite different. So I believe it would be interesting to see how they compare both in their model structure, and their performance on a few datasets in practice.

---

> ### Author Response · Authors · 2023-11-19
>
> Thank you for feedback and constructive comments. We updated the paper to meet your immediately actionable suggestions (omitted from the rebuttal). Please see below our responses to your comments:
>
> Weakness:
>
> 1. (& question 1) Upon your request, we included the results of the calibration modeling with GPs denoted as CMGP in the revised version of our paper and at the link below:
>
> [Adding CMGP](https://anonymous.4open.science/r/Corruption-resilient-Forecasting-Models-15E8/Additional-baselines.pdf)
>
> 2. When referring to observations, we include variables, including the target and co-variates, that have been observed in the past. In Figure 2, we provide an example where observations $X$ consist of of $X = \{X_1, X_2, \gamma\}$.

---

> > ### Comment · Reviewer_zFSw · 2023-11-19
> >
> > I thank the authors for the added baseline using the Gaussian process.

---

### Official Review · Reviewer_Hpfd · 2023-10-30

**Soundness:** 3 good
**Presentation:** 2 fair
**Contribution:** 2 fair
**Rating:** 6
**Confidence:** 4

**Summary:**

The paper proposes a novel joint forecast-corrupt-denoise model that consists of the forecasting module and the corrupt-denoising module. The forecasting model focuses on accurately predicting coarse-grained behavior. The corrupt-denoising model focuses on capturing fine-grained behavior, with a GP model employed to enforce the smoothness and co-relationship in added noise.

Empirical evaluations show the flexibility of the proposed framework in incorporating popular time-series forecasting models such as Informer and Autoformer and exhibit outperformance than popular time-series forecasting models.

**Strengths:**

1. The idea and framework introduced by the paper are natural and easy to follow.

2. The proposed framework adds a corrupt-denoising model as a tail of a forecasting model, which shows flexibility to incorporate with existing SOTA methods, and may further enhance their performance

3. The empirical evaluations show benefits of the proposed framework than single forecasting models alone. In addition, evaluations also illustrate the effectiveness of denoising GP corrupted time series than isotropic Gaussian noised time series

**Weaknesses:**

1.  Despite that the soundness of the methodology makes sense and is easy to follow, there are points remaining unclear more interpretations should be made. For instance, while it is understandable that adding noise through a Gaussian process can result in smoother and more structured noise patterns compared to isotropic Gaussian noise, it is still not that obvious which is worth formal illustrations from either intuitions or equations to explain the difference/benefits between adding isotropic Gaussian noise.

2. The effectiveness of the proposed framework is marginal when considering the framework doubles the parameter of a single forecasting model, e.g., the denoising model follows the forecasting model's architecture. In this case, it is also worthwhile showing that the benefits of the proposed framework are indeed by its mechanism, instead of overparameterization for better capability with a stack of Informers for example.

3. The ablation study is not well-exhibited. Despite there being a simple ablation study for the synthetic data in the Introduction, we expect to see some real-world examples or case studies to prove the statement 'The forecasting model focuses on accurately predicting coarse-grained behavior. The corrupt-denoising model focuses on capturing fine-grained behavior, with a GP model employed to enforce the smoothness and co-relationship in added noise', rather than simply the synthetic data.

4. The presentation needs to be improved. For example, there are repeated results in Table 1 and Table 2, which should be merged. And if MSE is the only metric used for evaluation, why is it necessary to assign it a column in all tables

**Questions:**

1. For results on DLinear, DeepAR, and ARIMA, why the reported variances are zeros?

2. Is the reported error, or ground truth normalized?

3. Does adding the denoising part to the forecasting model help or harm the convergence?

4. Does adding the denoising part to the forecasting model tend to lead to overfitting than the direct output of the forecasting model, say fine-grained behavior found by the denoising model overfits the ground truth when the ground truth is smooth? It might be measurable by case study or by using correlation metrics.

---

> ### Author Response · Authors · 2023-11-19
>
> Thank you for feedback and constructive comments. We updated the paper to meet your immediately actionable suggestions (omitted from the rebuttal). Please see below our responses to your comments:
>
> Weakness:
>
> 1. The success of our proposed model stems from the division of labors between the first forecasting stage (for coarse-grained) and the denoising stage (fine-grained predictions), where initial forecasts are blurred locally by utilizing a GP model. In between the parameters of the GP model are trained for best end-to-end performance.
>
> 2. We include the results of forecasting models with higher number of layers (parameters) in the Appendix and the following link, to show that higher number of parameters does not necessarily result in a better performance due to potential overfitting.
>
> [Results of standalone forecasting models with higher number of layers (parameters)](https://anonymous.4open.science/r/Corruption-resilient-Forecasting-Models-15E8/Additional-results-higher-number-parameters.pdf)
>
> 3. To meet your request we included the actual forecasts in the paper's Appendix and the provided link.
>
> [Actual Forecasts](https://anonymous.4open.science/r/Corruption-resilient-Forecasting-Models-15E8/Actual-forecasts-including-AutoDWC.pdf)
>
> Questions:
> 1. The standard errors for the mentioned models are less than 0.001 and we round the errors to three digits.
>
> 2. Ground-truth is zero mean normalized, and reported errors are further normalized using the mean of the absolute values of the ground truth data as a scaling factor.
>
> 3. (& 4) Our forecast-corrupt-denoise framework with GPs contributes to enhanced convergence and showcases a lack of overfitting as the performance on the hold-out test still improves. This is illustrated in the figures provided in the following link and the paper's Appendix section.
>
> [Convergence during training and validation](https://anonymous.4open.science/r/Corruption-resilient-Forecasting-Models-15E8/Convergence.pdf)

---

> > ### Comment · Reviewer_Hpfd · 2023-11-20
> > **Response to authors' comment**
> >
> > I thank the author's response and additional experiments to my questions.
> >
> > From my personal perspective, I believe the idea of adding noise and denoising processes in time-series forecasting is a novel and interesting idea. However, as shown in the empirical part, the improvement of the proposed method is kind of marginal, for most cases, the improvement between with and without DG is less than 10% lower in testing error. My worry is that when incorporating newer models such as Fedformer or PatchTST, the improvement from DG will become even marginal. (If it constantly shows an ~10% improvement, then I'd believe the proposed method is fairly good enough)
> >
> > One more suggestion is for the description of dataset and baseline model, the authors do not necessarily use paragraph headers for each. Instead, the author could use an Appendix for the detailed explanation, and save the space for more experiment results.

---

> > > ### Author Response · Authors · 2023-11-22
> > >
> > > Thank you for your response, upon your request and the time limit, we included the results of four treatments of the Fedformer model
> > > when predicting for 96 future time steps in terms of mean of std of MSE. We will continue to run the experiments for other forecasting horizons and include the results in the camera-ready version.
> > >
> > > Dataset| FedGP(Ours) | Fedformer | FedDI | FedDWC|
> > > | -------- | -------- | -------- | -------- | -------- |
> > > |  Traffic  |  $\boldsymbol{0.376} \pm 0.001$  |  0.391 $\pm 0.004$ | 0.388 $\pm 0.007$ | 0.393 $\pm 0.008$|
> > > |  Electricity   |  $\boldsymbol{0.267} \pm 0.003$ |  0.290 $\pm 0.001$ | 0.280 $\pm 0.005$ | 0.277 $\pm 0.001$|
> > > | Solar   |  $\boldsymbol{0.683} \pm 0.003$  |  0.737 $\pm 0.013$  | 0.727 $\pm 0.004$ | 0.725 $\pm 0.001$ |

---

> > > > ### Comment · Reviewer_Hpfd · 2023-11-22
> > > > **Response to authors' comment**
> > > >
> > > > I thank the reviewer again for the additional results. The results look good to me.

---

### Author Response · Authors · 2023-11-19

We thank all the reviewers for their thoughtful comments. We have prepared a revised paper draft based on the comments by including actual forecasts of our datasets, convergence plots, new baselines including CMGP, and an ablation study for higher number of layers (parameters) of standalone treated forecasting models. To adhere to the 9-page limit, all supplementary materials, except baseline CMGP, have been moved to Appendix section.

---

### Meta-Review · Area_Chair_RyaA · 2023-12-05

**Metareview:**

This work was reviewed by four reviewers. They found the paper well-motivated, with comprehensive studies and a novel approach to time-series forecasting. However, the reviewers also had several concerns: the paper lacks clarity in explaining the benefits of using Gaussian processes and its effectiveness seems marginal due to increased parameters. The ablation study was found inadequate, and the presentation needs refinement, with repetitive data and unclear metrics usage. The novelty is also questionable, as the framework resembles existing statistical models. Despite promising results, the paper needs better clarification on its methodology and distinctiveness. Thus, the reviewer consensus is to reject the work in its current form.

I recommend that you pay attention to the comments related to clarity and presentation before resubmitting this work.

**Justification For Why Not Higher Score:**

The paper has several issues.

**Justification For Why Not Lower Score:**

N/A

---

### Decision · Program_Chairs · 2024-01-16

Reject